

# Validation and measurement invariance of the Chinese version of the academic self-efficacy scale for university students

Mengyuan Zhao[1,2], Garry Kuan[2], Vinh Huy Chau[3] and Yee Cheng Kueh[4]

[1] Department of Sports Rehabilitation, School of Humanistic Medicine, Anhui Medical University, Hefei, China
[2] Exercise and Sports Science Programme, School of Health Sciences, Universiti Sains Malaysia, Kubang Kerian, Kelantan, Malaysia
[3] Ho Chi Minh City University of Physical Education and Sport, Ho Chi Minh City, Vietnam
[4] Biostatistics and Research Methodology Unit, School of Medical Sciences, Universiti Sains Malaysia, Kubang Kerian, Kelantan, Malaysia

## ABSTRACT

**Objectives**. The objective of this study was to examine the psychometric properties of the Chinese version of the Academic Self-Efficacy Scale (ASES-C) and confirm its measurement invariance across gender identities.

**Methods**. In this study, 502 university students (29.68% male, 70.32% female) with a mean age of 19.93 years (SD = 1.64) voluntarily participated. The Academic Self-Efficacy Scale (ASE) was utilized as a unidimensional measure of students' learning efficacy. The English version of ASES was translated into Chinese using a forward-backward translation procedure. Confirmatory factor analysis (CFA) and invariance testing were conducted with the single-factor model of ASES. Composite reliability (CR) and internal consistency were calculated based on Cronbach's alpha.

**Results**. Upon re-specification of the model, CFA results for the hypothesized single-factor model with eight items indicated an acceptable fit (CFI = 0.959, TLI = 0.943, SRMR = 0.036, RMSEA = 0.065). Cronbach's alpha and CR values were 0.785 and 0.880, respectively. Multi-group CFA results demonstrated measurement equivalence for the Chinese version of ASES across gender identities. The findings supported the measurement invariance of ASES-C for both male and female participants.

**Conclusion**. The ASES-C, consisting of one factor and eight items, is a reliable instrument for assessing Chinese university students' self-efficacy in learning. Furthermore, it is suitable for making meaningful comparisons across gender identities.

## INTRODUCTION

In the field of education, academic self-efficacy is widely recognised as one of the crucial factors influencing students' academic achievement and self-development (*Chen et al., 2023*). Academic self-efficacy refers to an individual's confidence and ability assessment in successfully completing academic tasks and achieving success (*Dixon, Hawe & Hamilton, 2020*). By enhancing students' academic self-efficacy, their motivation to learn can be

Corresponding author
Garry Kuan, garry@usm.my

stimulated, their academic performance can be strengthened, and their success in the academic domain can be promoted (*Ersanlı, 2015*). Conversely, students with decreased academic self-efficacy may be more susceptible to emotional distress (such as depression and anxiety) and exhibit more self-limiting behaviours (such as avoidance, low self-control, and diminished future-oriented behaviours) (*Honicke & Broadbent, 2016*).

In order to assess students' academic self-efficacy effectively, researchers *McIlroy, Bunting & Adamson (2000)* developed an academic self-efficacy scale (ASES), which has been widely utilized in academic research. This scale comprises a series of items that evaluate students' confidence and competence in academic tasks. By collecting students' self-evaluations, the scale enables the measurement of students' levels of academic self-efficacy with effectiveness.

However, despite the widespread application of the ASES in academic research (*Lavasani et al., 2011*; *Mirhosseini, Lavasani & Hejazi, 2018*; *Mirzaei-Alavijeh et al., 2018*), its applicability and validity in the Chinese context have not been adequately validated. Considering that different languages and cultural backgrounds may introduce variations in the understanding and expression of academic self-efficacy, it is necessary to conduct a confirmatory factor analysis of the Chinese version to ensure the accuracy and reliability of the scale in China or other Chinese-speaking regions.

Therefore, the purpose of this study was to conduct a confirmatory factor analysis of the Academic Self-efficacy Scale developed by *McIlroy, Bunting & Adamson (2000)* in Chinese university students. We aimed to further understand the structure and characteristics of students' academic self-efficacy in the Chinese context and validate the effectiveness and reliability of the scale in the Chinese cultural background. Additionally, the measurement invariance test of the questionnaire ensures that comparisons between different participant groups are both meaningful and effective. Without measurement invariance, observed intergroup differences may be influenced by measurement bias, leading to inaccurate conclusions (*Toth-Kiraly & Neff, 2021*). In particular, this study (1) employed confirmatory factor analysis (CFA) to validate the structural validity of the Academic Self-Efficacy Scale (ASES-C) and (2) once the ASES-C was established, verified its measurement invariance across male and female samples. It hypothesised that the Chinese version of ASES demonstrated satisfied structural validity and measurement invariance across different genders.

## MATERIAL AND METHODS

### Participants

A total of $N = 502$ university students participated in this study, with a majority of female participants (70.32%). The average age of the participants was 19.93 years (SD = 1.64). All participants were Chinese. According to *Kyriazos (2018)*, conducting a CFA requires a sample size of over 300, and the 500 samples in this study met this requirement.

### Measures

The academic self-efficacy scale developed by *McIlroy, Bunting & Adamson (2000)* aims to reflect *Bandura*'s (*1986*) definition of self-efficacy, including the initial determination to take action, the effort exerted during action, and the perseverance in the face of obstacles.

The scale was adapted by the authors from scales used for assessment in statistics at the University of Ulster. It consists of 10 items that measure the strength of students' expectations/beliefs regarding academic performance. Participants respond to the items using a 7-point Likert scale ranging from "Strongly Agree" to "Strongly Disagree." Seven items (including items 1, 2, 3, 4, 7, 8, and 10) require reverse scoring, and the final scores are obtained by reverse scoring. The items are then summed to give a total score for the measure. Higher scores reflect higher levels of academic self-efficacy. Previous studies have shown that the ASES has good internal consistency, with a Cronbach's alpha of 0.85 in one study (*McIlroy et al., 2015*).

## Questionnaire translation

Permission to use and translate the ASES was obtained from the original authors. The translation process involved both forward and backward translation procedures into Chinese (*Chai et al., 2022*). Firstly, a bilingual individual familiar with the content performed a forward translation from English to Chinese. Then, another bilingual individual conducted a backward translation from Chinese to English. Subsequently, a small group consisting of four domain experts proficient in both languages reviewed the translated content in English and the back-translated content in Chinese, comparing each item with its corresponding item in the original English version. The group members were also asked to examine the content of ASES-C to ensure that the items were culturally and conceptually suitable for Chinese university students. The final version of ASES-C was pilot-tested with a sample of 10 university students. The students completed the ASES-C questionnaire and were asked to provide feedback on the wording and presentation of the questionnaire. The measurement results from the students were satisfactory, requiring no modifications.

## Data collection

A cross-sectional research design was employed to study the self-administered ASES-C questionnaire. Convenience sampling was used to recruit participants for the study. Data collection took place from May 2022 to July 2022 at Jiangsu Vocational College of Medicine, Jiangsu Province, China. This study obtained ethical approval from the (Universiti Sains Malaysia Human Research Ethics Committee) Human Research Ethics Committee (USM/JEPeM/22040240) and was conducted in accordance with the principles outlined in the Declaration of Helsinki.

During the data collection period, in various WeChat study groups, class QQ groups, and other social media platforms, posted recruitment surveys. Chinese university students who were native Chinese speakers and willing to complete the questionnaire were included in the study. All data were collected online, and no offline paper questionnaires were used. In addition to the content of ASES-C, the questionnaire also included an investigation of participants' demographic information, including age, gender, grade, and major.

Since the questionnaire was anonymous, participants' responses could not be matched to individuals. Participants were required to click the "Agree" button after reading the information to the participant and providing informed consent (online) in order to proceed

to answer the questionnaire. Submitting the completed questionnaire was considered as consenting to participate in the study by default. No incentives were provided to participants in this study. A total of 536 questionnaires were initially collected. According to the established exclusion criteria: the questionnaire responses showed suspicious regularity, incorrect answers to common sense question ("which direction does the sun rise?"), and excessively short completion times, 34 questionnaires were identified as suspicious, the data from the remaining 502 participants were included in the analysis. All the participants were obtained the online informed consent.

## Statistical analysis

The data analysis was conducted using Mplus 8.3. All data were complete without any missing values. Based on the Mardia multivariate skewness and kurtosis test, the assumption of multivariate normality was violated ($p < 0.05$). Therefore, robust to maximum likelihood estimator, MLM, was employed in the operations of confirmatory factor analysis and measurement invariance testing.

The hypothesised measurement model consisted of 1 factor and 10 observed variables. The fit indices selected to evaluate the measurement model included the Comparative Fit Index (CFI) and the Tucker and Lewis Index (TLI), with desired values greater than 0.9, the Root Mean Square Error of Approximation (RMSEA) was less than 0.08, the Standardized Root Mean Residual (SRMR) was under 0.05 (*Wu, 2011*). The Chi-square test statistic and its degrees of freedom (df) are also reported. The theoretically expected factor loadings should be greater than 0.4 (*DeVon et al., 2007*). Reliability testing was conducted using Cronbach's alpha coefficient and composite reliability (CR), with recommended values equal to or greater than 0.60 (*Hair et al., 2010*).

The study employed multiple-group confirmatory factor analysis to examine the measurement invariance of ASES-C in a sample of university students with diverse gender identities. Measurement invariance encompasses four aspects of equivalence (*Luong & Flake, 2023*; *Maassen et al., 2023*): (1) configural invariance, which tests whether the latent variables have the same structure across different groups. The configural invariance ensures that the assessment of the same latent constructs is similar across different groups. In this step, no restrictions are placed on any parameters; (2) metric invariance, which tests whether the factor loadings are equivalent across groups. Implementing metric invariance means that the strength of the relationship between observed variables and latent factors remains consistent across different groups. In this step, while maintaining factor structure invariance, constraints are applied to factor loadings; (3) scalar invariance, which tests whether the intercepts of the observed variables are equivalent across groups. If scalar invariance is not supported, this could bias comparisons between means. In this step, the intercept is further constrained in the metric invariance model; and (4) strict invariance, which tests whether the error variances are equivalent across different groups. The violation of strict invariance indicates that measurement errors may vary between groups, potentially affecting group comparisons. In this step, further constraints on error variance are imposed on the scalar invariance model. Multiple-group confirmatory factor analysis involves a series of nested models, progressively moving from lenient to strict conditions to seek invariance.

**Table 1  The demographic characteristics ($n$ = 502).**

| Characteristics | Frequency | Percentage | Mean ± SD |
|---|---|---|---|
| Gender | | | |
| Male | 149 | 29.68% | |
| Female | 353 | 70.32% | |
| Age (years) | | | 19.93 ± 1.64 |
| Major | | | |
| Science and Engineering | 31 | 6.18% | |
| Medicine | 393 | 78.29% | |
| Economics | 12 | 2.39% | |
| Management | 16 | 3.19% | |
| Law | 8 | 1.59% | |
| Pedagogy | 42 | 8.36% | |
| Class | | | |
| Freshman | 272 | 54.18% | |
| Sophomore | 176 | 35.06% | |
| Junior | 28 | 5.58% | |
| Senior | 19 | 3.78% | |
| Postgraduate | 7 | 1.40% | |

In this study, $\Delta$CFI, $\Delta$TLI, and $\Delta$RMSEA were used as evaluation criteria for measurement invariance. If $\Delta$CFI and $\Delta$TLI are less than or equal to 0.01 and $\Delta$RMSEA is less than 0.015, the measurement invariance is considered acceptable (*Cheung & Rensvold, 2002*). Additionally, after establishing measurement invariance for the ASES-C, an independent samples $t$-test was used to compare the differences in academic self-efficacy between males and females.

# RESULTS

The majority of participants consisted of first-year and second-year university students, with a predominance of female participants. Specific information and other descriptive statistics are presented in Table 1.

The mean scores of each item on the scale ranged from 4.06 to 5.13. After removing items 6 or 9 from the ASES, the Cronbach's alpha values were 0.837 and 0.84, respectively, which were higher than the Cronbach's alpha value after removing other items. The mean, standard deviation, skewness coefficient, kurtosis coefficient, and Cronbach's alpha value after excluding items are shown in Table 2.

## Measurement model for the ASES-C

The hypothesised measurement model of ASES-C consisted of 10 items representing a single-dimensional factor. The fit indices of the baseline model (Model 1) did not meet the standard criteria, indicating an inadequate fit due to low factor loadings on certain items. By removing ASES 6 and ASES 9, Model 2 was obtained, which demonstrated acceptable fit indices across all measures. Based on Model 2, the calculated Cronbach's alpha was 0.785, and the composite reliability was 0.880, both exceeding the reference values. Specific fit

**Table 2   Means, SDs, skewness, kurtosis and Cronbach's alpha if deleted results ($n = 502$).**

| Item | Mean | Standard deviation | Skewness | Kurtosis | Cronbach's alpha if deleted |
|---|---|---|---|---|---|
| ASES 1 | 4.94 | 1.211 | 0.115 | −0.105 | 0.762 |
| ASES 2 | 4.53 | 1.173 | 0.381 | 0.575 | 0.759 |
| ASES 3 | 4.57 | 1.232 | 0.345 | 0.196 | 0.761 |
| ASES 4 | 4.82 | 1.178 | 0.257 | −0.142 | 0.753 |
| ASES 5 | 4.4 | 1.177 | 0.332 | 0.611 | 0.796 |
| ASES 6 | 4.06 | 1.195 | −0.102 | 0.89 | 0.837 |
| ASES 7 | 4.78 | 1.164 | 0.328 | 0.094 | 0.751 |
| ASES 8 | 5.13 | 1.229 | 0.16 | −0.956 | 0.76 |
| ASES 9 | 4.48 | 1.308 | −0.067 | 0.397 | 0.84 |
| ASES 10 | 4.39 | 1.203 | 0.334 | 0.746 | 0.79 |

**Table 3   Goodness of fit indices of the tested measurement models.**

| Path models | $\chi^2$(df) | RMSEA (90% CI) | CFI | TLI | SRMR |
|---|---|---|---|---|---|
| Model 1 | 195.388 (35) | 0.096 (0.083–0.109) | 0.863 | 0.823 | 0.079 |
| Model 2[a] | 62.532 (20) | 0.065 (0.047–0.084) | 0.959 | 0.943 | 0.036 |

**Notes.**
[a] Measurement model with items deleted (ASES 6 and ASES 9).
RMSEA, root mean square error of approximation; CFI, comparative fit indices; TLI, Tucker and Lewis index; SRMR, standardized root mean square.

**Table 4   Standardized factor loadings for Model 1, and Model 2 of the ASES-C.**

| Items | Factor Loadings | | Cronbach's alpha (Model 2) | Composite Reliability (Model 2) |
|---|---|---|---|---|
| | Model 1 | Model 2 | 0.785 | 0.880 |
| ASES 1 | 0.713 | 0.712 | | |
| ASES 2 | 0.763 | 0.763 | | |
| ASES 3 | 0.750 | 0.751 | | |
| ASES 4 | 0.814 | 0.814 | | |
| ASES 5 | 0.434 | 0.437 | | |
| ASES 6 | 0.091 | – | | |
| ASES 7 | 0.827 | 0.826 | | |
| ASES 8 | 0.693 | 0.691 | | |
| ASES 9 | 0.025 | – | | |
| ASES 10 | 0.489 | 0.490 | | |

indices for each model are presented in Table 3, and factor loadings for different models can be found in Table 4.

## Measurement invariance

From Table 5, it can be observed that when comparing the weak invariance model to the configural invariance model, and the strong invariance model to the weak invariance model, $\Delta$CFI $\leq$ 0.010, $\Delta$TLI $\leq$ 0.010, and $\Delta$RMSEA <0.015. However, when comparing the

**Table 5   Measurement invariance testing results of the ASES-C across genders (*n* = 502).**

| Model | $\chi^2$(df) | CFI | TLI | RMSEA (90% CI) | ΔCFI | ΔTLI | ΔRMSEA |
|---|---|---|---|---|---|---|---|
| Configural | 92.151 (40) | 0.952 | 0.933 | 0.072 (0.053–0.091) | – | – | |
| Metric | 100.962 (47) | 0.950 | 0.941 | 0.068 (0.049–0.086) | −0.002 | 0.008 | −0.004 |
| Scalar | 115.496 (54) | 0.943 | 0.941 | 0.067 (0.050–0.084) | −0.007 | 0 | −0.001 |
| Strict | 111.763 (62) | 0.954 | 0.959 | 0.057 (0.039–0.073) | 0.011 | 0.018 | −0.010 |

Notes.

$\chi^2$, chi-square goodness of fit; df, degrees of freedom; CFI, Compartative Fit Index; TLI, Tucker–Lewis index; RMSEA, root mean square error of approximation; 90% CI, 90% confidence intervals; ΔCFI, CFI difference; ΔTLI, TLI difference; ΔRMSEA, RMSEA difference.

**Table 6   Sex differences in the average item and total scores of ASES-C.**

| Items | Males (*n* = 149) | Females (*n* = 353) | 95% CI | *t* | *p* |
|---|---|---|---|---|---|
| ASES 1 | 5.05 (1.25) | 4.9 (1.19) | [−0.08–0.384] | 1.284 | 0.200 |
| ASES 2 | 4.7 (1.25) | 4.46 (1.13) | [0.009–0.458] | 2.044 | 0.042[*] |
| ASES 3 | 4.6 (1.30) | 4.56 (1.20) | [−0.191–0.283] | 0.381 | 0.703 |
| ASES 4 | 4.88 (1.26) | 4.79 (1.14) | [−0.137–0.315] | 0.772 | 0.441 |
| ASES 5 | 4.53 (1.27) | 4.35 (1.13) | [−0.041–0.41] | 1.607 | 0.109 |
| ASES 7 | 4.81 (1.25) | 4.76 (1.13) | [−0.174–0.274] | 0.44 | 0.660 |
| ASES 8 | 5.07 (1.23) | 5.15 (1.23) | [−0.315–0.157] | −0.659 | 0.510 |
| ASES 10 | 4.55 (1.22) | 4.32 (1.19) | [0.003–0.463] | 1.989 | 0.047[*] |
| Total Score | 38.19 (7.84) | 37.29 (6.58) | [−0.532–2.349] | 1.242 | 0.215 |

Notes.

[*]$p < 0.05$.

CI, Confidence Interval.

strong invariance model to the strict invariance model, ΔCFI and ΔTLI slightly exceeded the recommended critical values, but ΔRMSEA was −0.010, which is smaller than the critical value.

## Comparison of academic self-efficacy between male and female students

From Table 6, only ASES 2 and ASES 10 exhibited statistically significant differences between male and female students. For other items and the overall scale score, there were no statistically significant differences. Based on this assessment tool's overall scores, we could conclude that academic self-efficacy did not differ significantly between male and female students in the present sample.

## DISCUSSION

The purpose of this study was to assess the structural validity and reliability of the Chinese version of ASES. Additionally, the study also evaluated the measurement invariance of ASES-C. This study provides evidence for the adequate psychometric properties of ASES-C.

The CFA results of this study provided support for the unifactorial model of ASES-C. The sample data from different gender groups supported the single-factor structure, consistent with the original research findings (*McIlroy, Bunting & Adamson, 2000*). In the

ASES-C model with eight items, all factor loadings were above 0.4, suggesting satisfactory structural validity (*Ibrahim et al., 2021*; *Kuan et al., 2019*). Based on the final model, the Cronbach's alpha value was 0.785, and the composite reliability value was 0.880, indicating good reliability of the Chinese version of ASES.

There are several reasons that may lead to low factor loadings for the item 6 and 9 in the scale. One possible reason is the cultural difference between China and the West. In Chinese culture, students may exhibit higher academic pressure and a stronger collectivist tendency (*Tan et al., 2021*), which could influence their responses to these items. The term 'fear' in ASES 9 might be understood as an extremely negative emotion in the Chinese context, rather than a general academic anxiety. The ASES 6 and ASES 9 are both negatively phrased items in the scale. In survey design, negative items could sometimes confuse or unsettle respondents, especially when they are accustomed to positively framed statements. This inconsistency in responses may lead to reduced factor loadings. Additionally, students may tend to provide lower levels of agreement when responding to negatively phrased items due to social expectations or personal image maintenance. This could impact the factor loadings of the items. Future research could explore this pattern further by examining student samples from different regions and schools to determine whether this pattern can be replicated.

The significance of studying academic self-efficacy among university students lies in uncovering and understanding individual differences within the academic domain and their impact on learning achievement and academic development. However, understanding individual differences relies on measurement tools that exhibit measurement invariance across different populations (*Dong & Dumas, 2020*; *Kueh et al., 2021*). In this study, the measurement invariance of ASES-C was found to be satisfactory across genders. This indicated that male and female university students had a similar understanding of all items in ASES-C. This equivalent understanding is necessary for accurately comparing the differences in academic self-efficacy between male and female students.

In the nested model equivalence tests, the results demonstrated that ASES-C supports full configural invariance, indicating that the latent variable structure is the same across different samples of male and female students. Additionally, the weak invariance test results were met, indicating that the factor loadings of the Chinese version of ASES are equivalent across male and females. Furthermore, the strong invariance test results satisfied the measurement criteria, suggesting that the ASES-C has the same intercepts across different genders. Lastly, the strict invariance of the ASES-C was examined between male and female university students. Although not all three index change values were smaller than the critical values, previous literatures (*Sabo et al., 2022*; *Swami et al., 2023*; *Wang & Bi, 2018*) have suggested that when using ΔCFI and ΔTLI values to compare nested models, values smaller than 0.01 indicate good fit of the nested model, accepting the measurement invariance model. Values between 0.010 and 0.020 indicate a moderate worsening of fit, which cannot determine the presence or significance of differences. Values greater than 0.020 indicate a definite difference. However, overall, the worsening of fit was not significant, and it can be considered that strict invariance holds, indicating that the error variances are equal between the male and female samples.

This study is not without limitations. Firstly, caution should be exercised in interpreting these results as the data were collected from a single university, which may affect the generalizability of the findings. Secondly, the use of self-report measures to assess the research outcomes may introduce response biases and potentially decrease the accuracy of the collected data, which is an inherent limitation of self-report questionnaire data collection. However, we ensured the confidentiality of participant data and encouraged them to respond to all items truthfully. Lastly, while the content validity and reliability were extensively examined, this study primarily focused on the structural validity of ASES-C through CFA. The extent of validity in other aspects of ASES-C remains unknown, and future research could incorporate additional validity and reliability tests, such as concurrent, predictive, convergent, and discriminant validity, and also test-retest reliability, to gain a more comprehensive understanding of the effectiveness of ASES-C and establish a more robust foundation for its application.

## CONCLUSION

In this study, ASES-C has been demonstrated to have sufficient reliability and validity for assessing academic self-efficacy among Chinese university students. Furthermore, the scale has shown adequate measurement invariance (configural, weak, strong, and strict) across different genders. These findings support the use of ASES-C for reliable quantitative comparisons between males and females.

### Funding
This research was supported by the Ministry of Higher Education Malaysia for Fundamental Research Grant Scheme (FRGS) with Project Code: FRGS/1/2020/SKK06/USM/03/13. The funders had no role in study design, data collection and analysis, decision to publish, or preparation of the manuscript.

### Grant Disclosures
The following grant information was disclosed by the authors:
The Ministry of Higher Education Malaysia for Fundamental Research Grant Scheme (FRGS): FRGS/1/2020/SKK06/USM/03/13.

### Competing Interests
The authors declare there are no competing interests.

### Author Contributions
- Mengyuan Zhao conceived and designed the experiments, performed the experiments, analyzed the data, prepared figures and/or tables, authored or reviewed drafts of the article, and approved the final draft.
- Garry Kuan conceived and designed the experiments, performed the experiments, authored or reviewed drafts of the article, and approved the final draft.

- Vinh Huy Chau analyzed the data, prepared figures and/or tables, authored or reviewed drafts of the article, and approved the final draft.
- Yee Cheng Kueh analyzed the data, authored or reviewed drafts of the article, and approved the final draft.

## Human Ethics

The following information was supplied relating to ethical approvals (*i.e.,* approving body and any reference numbers):

This study obtained ethical approval from the Universiti Sains Malaysia Human Research Ethics Committee (USM/JEPeM/22040240) and was conducted in accordance with the principles outlined in the Declaration of Helsinki.

## Data Availability

The raw data is available in the Supplemental File.

## Supplemental Information

Supplemental information for this article can be found online at http://dx.doi.org/10.7717/peerj.17798#supplemental-information.

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
