# Peer review of "Validation and measurement invariance of the Chinese version of the academic self-efficacy scale for university students"

_PeerJ, doi:10.7717/peerj.17798_

## Round 0.1 · original submission · Major Revisions

Dear Authors,

I have received the reviewers' comments. Having read the manuscript myself, I agree with the concerns raised by R2 and would encourage you to address them. Moreover, I would like you to address also the minor details highlighted by R1.

Looking forward to receiving your response, kind regards

·

Basic reporting

The basic reporting is generally of an acceptable standard and written with good, clear style. One exception to this is referencing and I have highlighted this in Addition comments below.

Experimental design

The design for the study is appropriate with commendable application of appropriate statistics.

Validity of the findings

The findings are valid and offer an original application of the measure with potential for extending its application across culture.

My more detailed comments can be found in Additional comments.

Additional comments

Validation and measurement invariance of the Chinese version of the academic self-efficacy scale for university students (#95522)

This is a competent and coherent article that translates and validates the ASES in Chinese. The authors have provided a good, basic summary of the usefulness of the measure and the value of the measure in Chinese. This will potentially open the door to much wider use of the measure and this provides a solid rationale for the exercise.

Preliminary work was undertaken in consulting with students as the end users of the measure and this served to augment the clarity, relevance and adequacy of the measure. Care was also taken before this in forward and backward translation to provide as far as possible accuracy in representing the original concepts.

Sound use of statistical methods included factor loadings, model fit indicators and application of invariance testing. These and criteria associated with them as cut off points are supported with authoritative sources and the findings are reported with due caution. The omission of two items from the original measure are warranted based on model fit and factor loadings.

The use of the literature gave a flavour of how useful the concept and measurement of academic self-efficacy is and highlight the value to its application to a range of educational concepts and challenges.

I am happy to recommend this article for publication with a few suggestions for minor modifications:

On page 4 of the manuscript under Participants, add to 502: N = 502.

On page 5, line 80. After the sentence ending in “obtained by reverse scoring”, add the sentence, “The items are then summed to give a total score for the measure”.

On page 6, line 125, the researchers names should be capitalised (first letter) and the conjunction and should be removed. Should be the Tucker Lewis index (TLI).

On page 9, line 189, the use of Transgender has a specific meaning in Western culture and could be misinterpreted so the expression would be better just as “are equivalent across male and females”.

On page 10, line 209, after criterion-related validity, it would be useful to add, “and predictive validity”.

I agree that it was the right decision based on the statistics to omit the two items (ASES 6 and ASES 9) - page 7, line 151.

Do the authors have any suggestion with reference to the content on why these two items did not load acceptably with the rest? It not they should at least note that this should be further explored in other Asian samples to see if this pattern is replicated.

References

 If APA referencing is to be followed then there are some violations within the list of References (although not generally in the text) that should be corrected.

 One exception to the above in the text is when the list does not follow alphabetical order – e.g. on page 4, line 53 where Lavansani should be before the other two.

 Journal titles should be italicised – e.g. the first reference should be – Journal of Social and Clinical Psychology

 The year of publication should be bracketed (e.g. again in the first reference) – Bandura, A. (1986).

 Main words in journal titles should be capitalised – lines 250 and 251 – Personality and Individual Differences = please apply throughout the reference list.

 Add the ampersand (&) before the last name in the list where there is more than one name, as well as a comma after family name and a full stop after initials – e.g. line 249. Dong, Y. & Dumas, D. (2020).

Reviewer 2 ·

Basic reporting

I have had the pleasure of reviewing your manuscript entitled “Validation and measurement invariance of the Chinese version of the academic self-efficacy scale for university students.”
The study aimed to examine the structural validity and internal consistency of the Chinese version of the Academic Self-Efficacy Scale and confirm its measurement invariance across gender identities. Overall, I liked your transparent approach, and I see the relevance of your data to the field of psychological measurement. The manuscript is also well-structured and written.

The rationale for testing invariance across different groups should be clearly outlined in your introduction.

Including the results of further analyses based on your existing data would also be enriching. For example, it would be intriguing to explore whether self-efficacy mean scores vary by gender. Similarly, it would be interesting to see if there are differences according to another socio-demographic indicator, namely between first-year students and those in their second year and beyond. First-year students might assess their self-efficacy differently from upper-year students because they are less familiar with the academic context and activities. To support this analysis, you could test the invariance of the measure for these groups, thereby enriching the information on the structural validity of the scale.

Regarding statistical details, providing descriptive statistics for all scale items (mean, standard deviation, skewness, kurtosis) would be beneficial. Additionally, you might consider including other item-related indices such as corrected item-total correlation and Alpha if the item is Dropped. All of these could provide a more nuanced picture of the results and help clarify why certain items did not perform as expected.

Experimental design

The study design is simple yet effective, aligning well with the stated objectives. However, the nomological pattern of academic self-efficacy was not investigated. This would provide important and crucial information on the measure's validity, such as concurrent, predictive, convergent, and discriminant validity. The temporal stability of the measure (e.g., test-retest) was also not explored. I believe these aspects should be emphasized more strongly within the limitations section. Ideally, a second study would be conducted to analyze these additional aspects. A follow-up study might be challenging but could significantly improve the knowledge of the scale psychometric properties.

While your methodology is generally well-described and transparent, greater detail on your sampling method and the recruitment process (e.g., via social media, among students attending a specific course) would clarify your approach.
The statement that "34 questionnaires were potentially randomly completed" needs clearer elaboration on how this was concluded.
The description of the invariance analysis accurately reports the various dimensions of invariance, yet it remains unclear how the different models are tested against each other and what the implications are of not achieving a certain level of invariance (for example, if scalar invariance is not supported, this could bias comparisons between means). This is significant because providing more details on these aspects could be greatly helpful to those unfamiliar with such analysis, which is currently underutilized (https://doi.org/10.1037/met0000624).
The methodology for calculating the total score of academic self-efficacy is not described. Doing this would prevent any potential confusion and aid in the practical application of your research

Validity of the findings

In lines 167-175 of the discussion, the repetition of results should be revised to emphasize the evidence on structural validity and internal consistency of the scale without redundancy.
Given the mixed results regarding strict invariance, I would be cautious in asserting that it is indeed upheld.
The data from this study are openly available. To enhance the interoperability of data sharing (FAIR principles; https://doi.org/10.1038/sdata.2016.18), I recommend making the data available in open formats (e.g., CSV).

Additional comments

Thank you for the opportunity to review your work. I look forward to your revised manuscript.

---

## Round 0.2 · accepted · Accept

Dear Authors,

Thank you so much for carefully addressing the comments of the Reviewers, they were very pleased for your response.

Reviewer 2 ·

Basic reporting

No comment

Experimental design

No comment

Validity of the findings

No comment

Additional comments

Dear Authors,
Thank you for submitting the revised version of your manuscript entitled “Validation and measurement invariance of the Chinese version of the academic self-efficacy scale for university students.” I have carefully reviewed the changes and am pleased with how you have addressed the various comments and suggestions provided in the initial review.
I appreciate the effort you have put into revising your work, and I am confident that these improvements will be well-received by the readers. I look forward to seeing your article published soon.
All the best